# A Compact Real-Time PCR System for Point-of-Care Detection Using a PCB-Based Disposable Chip and Open-Platform CMOS Camera

**DOI:** 10.3390/s25103159

**Published:** 2025-05-17

**Authors:** MinGin Kim, Sung-Hun Yun, Sun-Hee Kim, Jong-Dae Kim

**Affiliations:** 1Thermo Fisher Scientific, South San Francisco, CA 94080, USA; hellingford@gmail.com; 2School of Software, Hallym University, Chuncheon-si 24252, Republic of Korea; butter9709@gmail.com; 3Department of Fashion Industry, Incheon National University, Incheon 22012, Republic of Korea

**Keywords:** Real-time PCR, POCT, fluorescence detection, *Chlamydia trachomatis*, PCB-based chip, CMOS camera

## Abstract

We present a compact and cost-effective real-time PCR system designed for point-of-care testing (POCT), utilizing a PCB-based disposable chip and an open-platform CMOS camera. The system integrates precise thermal cycling with software-synchronized fluorescence detection and provides real-time analysis through a dedicated user interface. To minimize cost and complexity, a polycarbonate reaction chamber was integrated with a PCB-based heater and thermistor. A slanted LED illumination setup and an open-platform USB camera were employed for fluorescence imaging. Signal alignment was enhanced using device-specific region-of-interest (ROI) tracking based on copper pad corner detection. Thermal cycling performance achieved a heating rate of 8.0 °C/s and a cooling rate of −9.3 °C/s, with steady-state accuracy within ±0.1 °C. Fluorescence images exhibited high dynamic range without saturation, and the 3σ-based ROI correction method improved signal reliability. System performance was validated using *Chlamydia trachomatis* DNA standard (10^3^ copies), yielding consistent amplification curves with a Ct standard deviation below 0.3 cycles. These results demonstrate that the proposed system enables rapid, accurate, and reproducible nucleic acid detection, making it a strong candidate for field-deployable molecular diagnostics.

## 1. Introduction

Despite ongoing advances in medical science, infectious diseases continue to pose significant threats to global health and the economy [1,2,3,4,5]. High-risk pathogens such as Ebola, Zika, chikungunya, dengue, malaria, HIV, and the recent SARS-CoV-2 virus have shown the potential to cause global pandemics due to their high infectivity. Rapid diagnosis is essential for containing outbreaks and initiating appropriate treatment, and, therefore, sensitive and specific diagnostic tools play a critical role in infectious disease control [5,6].

Among the three major diagnostic methods—pathogen culturing, serological testing, and nucleic acid detection—PCR-based nucleic acid detection has emerged as the preferred technique. Compared to traditional culturing, PCR does not require highly trained personnel or complex lab infrastructure, and unlike serological methods, it does not rely on antibody synthesis [3,5,7,8]. Real-time PCR, in particular, provides high specificity by amplifying and detecting target nucleic acid sequences through temperature cycling and real-time fluorescence monitoring [2,9].

Conventional PCR protocols involve thermal cycling, followed by gel electrophoresis and image analysis of amplified products. These multi-step procedures increase the risk of cross-contamination during sample handling. To address this, real-time PCR systems that integrate fluorescence detection during amplification have been developed [10,11,12,13]. During the COVID-19 pandemic, the World Health Organization (WHO) recommended RT-PCR (real-time reverse transcription PCR) as the gold standard for virus detection [14].

However, deploying such systems in low-resource settings remains a challenge. Point-of-care testing (POCT) platforms have gained attention for enabling rapid, on-site diagnosis. Yet most commercial real-time PCR systems rely on expensive CCD sensors or photodiodes for high-sensitivity fluorescence detection. These systems often incorporate complex optical components—lenses, dichroic mirrors, and long optical paths—which limit their miniaturization and affordability [15,16,17]. As a result, real-time PCR systems are primarily confined to centralized laboratories, limiting their availability in urgent or remote scenarios [3,18,19,20,21]. In developing countries, the high cost of commercial PCR equipment further hinders adherence to WHO diagnostic standards [5].

To address these limitations, recent efforts have focused on developing POCT-compatible PCR platforms that are portable, affordable, and capable of delivering fast results from minimal sample volumes [3,22,23,24]. Many of these systems leverage microfluidics to enhance heat transfer and reduce reaction time [5,15,25,26]. In parallel, the use of smartphone cameras has been explored to lower the cost of fluorescence detection [27,28,29,30,31]. With recent advances in open-platform CMOS imaging modules, low-cost and compact alternatives to conventional CCD-based optical systems are becoming increasingly viable [32,33].

Our research group has previously introduced a cost-effective real-time PCR system using a PCB-based disposable chip [34,35]. The chip incorporates a heating element and thermistor directly on a PCB substrate, and the reaction chamber is fabricated from polycarbonate with sealing using off-the-shelf adhesive tapes. We have also demonstrated the feasibility of four-color multiplex detection using this platform [36].

In this study, we propose a compact and low-cost real-time PCR system optimized for point-of-care testing (POCT), based on a PCB-integrated disposable chip. The system integrates a microcontroller-driven thermal cycling module and an open-platform CMOS camera for fluorescence detection. A droplet-shaped polycarbonate reaction chamber was designed for efficient sealing and optical clarity, and the camera system employs slanted illumination to minimize optical path complexity. To enable reliable quantification, we optimized both the thermal control and image acquisition parameters. *Chlamydia trachomatis* (CT) standard DNA was used to validate the system. Experimental results showed accurate thermal control (±0.1 °C), high dynamic range in fluorescence imaging without saturation, and strong reproducibility (Ct variation < 0.3 cycles). These results demonstrate the system’s feasibility for POCT applications, especially in resource-limited environments. To implement this system, we developed a custom PCB-based chip, a compact thermal cycling module, and a synchronized fluorescence imaging pipeline, all described in the following sections.

## 2. Materials and Methods

### 2.1. PCR Chip Design

The PCR chip used in this study improves upon a previous design by significantly enhancing reagent injection and sealing, as illustrated in Figure 1 [36]. The reaction chamber, where DNA amplification and fluorescence detection occur, was fabricated from polycarbonate. Polycarbonate was selected for the chamber material due to its high heat resistance (deflection temperature > 130 °C), excellent optical clarity, and low autofluorescence, which are essential for fluorescence detection. It is biocompatible with PCR reagents, inexpensive, and easily moldable or laser-cut, making it ideal for mass production of disposable chips. The chamber was molded into a standalone polycarbonate part with a concave shape, including an inlet for reagent injection and an outlet for pressure balancing to facilitate smooth reagent filling.

To direct bubbles formed during thermal cycling toward the outlet, the chamber was shaped like a droplet. Sealing was achieved using qPCR-specific adhesive tape (9795R, 3 M, St. Paul, MI, USA), preventing rupture due to pressure increases from reagent expansion and vaporization.

The back side of the PCB includes a heating pattern for thermal cycling and an NTC (Negative Temperature Coefficient) thermistor (NCP15XH103D03RC, Murata Manufacturing Co., Ltd., Kyoto, Japan) for temperature sensing. The front side features a thermal spreading pad coated with tin to enhance fluorescence brightness gain. The heating pattern is composed of 100 μm-wide copper traces with 100 μm spacing, uniformly covering the chamber area. This geometry was determined empirically through iterative prototyping without numerical simulation, in order to ensure manufacturability on a low-cost, single-layer FR4 substrate while achieving sufficient thermal uniformity. The completed PCB is assembled with the lower housing using medical-grade double-sided adhesive tape (1510, 3 M). Both inlet and outlet are sealed using silicone rubber caps, which are manually pressed to close, as depicted in Figure 1.

The chip is designed as a disposable unit, primarily because both the PCB and thermistor are extremely inexpensive when mass-produced, making a single use economically viable. A single-layer FR4 PCB with an integrated copper heater and a low-cost thermistor can be fabricated for under $1 USD per unit at scale. This allows the entire chip to be discarded after each test, eliminating contamination risks without significantly increasing cost.

Additionally, disposability simplifies mechanical integration: the embedded heater and thermistor require consistent thermal contact, which would be difficult to maintain in a reusable configuration involving detachable connectors or alignment components. From a diagnostic perspective, single-use design also prevents carry-over contamination between samples, which is particularly important in field or home-use PCR settings where proper sterilization is not always feasible. Additional details of the molded polycarbonate chamber, including its groove depth and chamber opening dimensions, are provided in Appendix A.

### 2.2. Hardware Architecture

Figure 2 presents the functional block diagram of the complete system, which consists of two main subsystems: a thermal cycling system for PCR and an optical system for fluorescence detection.

The PCR system is controlled by a microcontroller (PCR MCU), implemented using a PIC18F4553 (Microchip Technology Inc., Arizona, TX, USA), which supports ADC (analog-to-digital conversion), PWM (pulse-width modulation), and USB interface functionalities. During experiments, the PCR chip is mounted into a dedicated chip connector. The heating element and fan are controlled via PWM signals to regulate temperature. Forced-air cooling with a compact axial fan (MR3010H05B-RSR, Mechatronics Fan Group, Preston, WA, USA) was selected instead of Peltier-based modules, in order to reduce system size, cost, and power consumption while still achieving sufficient cooling rates for PCR thermal cycling.

Temperature readings from the chip’s thermistor are obtained through a voltage divider circuit and converted using a Steinhart–Hart equation-based lookup table. Every 2 ms, ten ADC samples are averaged, and the resulting value is processed through a five-sample median filter to reduce noise.

The optical system comprises an excitation unit for LED control and an open-platform camera for fluorescence measurement. A separate microcontroller (Optic MCU, Trinket M0, Adafruit, New York, NY, USA) is employed to maximize PWM resolution and enable future expansion for filter wheel control and function separation. The excitation unit includes an LED and excitation filter; LED brightness and switching are managed via PWM signals. Fluorescence images are captured using a USB-compatible camera module with an integrated IMX298 CMOS sensor and autofocus lens (IMX298 Mini, Arducam, Kowloon, Hong Kong, China).

Both microcontrollers and the camera are connected to the host PC via a USB hub. A 5 V 3 A DC adapter was used to supply stable power to the heater, fan, and USB-connected modules. While a detailed power consumption analysis has not yet been conducted, the system operated reliably under this power envelope during all experiments. Since the current prototype is designed for benchtop evaluation, battery-based autonomous operation was not implemented at this stage.

Figure 3a illustrates the overall system architecture, and Figure 3b shows the assembled device. To achieve compactness and cost-efficiency, a slanted illumination configuration was adopted: the excitation LED is positioned at a 45° incidence angle relative to the PCR chamber. The camera is positioned at the chamber’s focal distance, with an emission filter mounted in front of the lens. All components and housing were fabricated using 3D printing. For fluorescence detection, a 470 nm blue LED (C503B-BAS-CY0C0461, Cree LED Inc., Durham, NC, USA) was used to excite FAM dye. The excitation and emission filters were selected to center around 470 nm (ET470/30x, Chroma Technology Corp., Bellows Falls, VT, USA) and 520 nm (ET520/20m, Chroma Technology, USA), respectively. Given the 5 mm LED and 3 mm camera lens diameters, 5 mm diameter, 1 mm thick optical filters were adopted.

### 2.3. Software Architecture

#### 2.3.1. Overall Architecture

Figure 4 shows the software block diagram of the overall system. The host PC runs two main software modules: the PCR Controller and the Optic Server.

The PCR Controller is developed in C/C++ using the Microsoft Foundation Class library to provide a graphical user interface (GUI) application. It manages the real-time PCR protocol loaded by the user and communicates with the PCR MCU via USB/HID to perform thermal cycling. Considering the timing requirements of real-time systems, user interface operations and protocol management tolerate latencies on the order of hundreds of milliseconds to seconds, whereas temperature control requires precise response times in the millisecond range. Therefore, a host-local architecture was implemented: high-level user interaction is handled by the host, while low-level timing-critical tasks are executed by the local system (PCR MCU).

The local system runs in a periodic loop, receiving commands from the host and reporting back its state. Since it only forwards received commands to peripheral devices and maintains status, this passive communication design simplifies the microcontroller logic and enables a compact and low-cost implementation. A schematic of the host-local architecture is provided in Appendix A.

When fluorescence detection is required during protocol execution, the PCR controller sends a TCP socket request to the Optic Server. The Optic Server is responsible for acquiring fluorescence images and calculating the relative fluorescence unit (RFU). It comprises a Shot Thread for RFU calculation, an Optic Controller for LED control, and a Camera Thread for continuously acquiring frames from the camera. The camera thread communicates with the open-platform IMX298 camera via the USB Video Class protocol.

The Optic Server is developed in Python 3.11.4 and uses the DirectShow and OpenCV libraries for camera setup and control. The LED’s brightness and switching are controlled using PWM signals from the Optic MCU.

#### 2.3.2. PCR Controller

Table 1 presents the real-time PCR protocol adopted in this study. The PCR controller processes the protocol row by row, where each row represents an action. The protocol begins with an initial denaturation step at 95 °C for 5 min (Label 1), followed by a typical thermal cycle consisting of denaturation (95 °C), annealing (55 °C), and extension (72 °C) stages (Labels 2–4). This cycle is repeated 35 times as indicated by the “GOTO” row, which jumps back to Label 2.

Fluorescence detection is typically performed at the end of the annealing or extension steps. This is triggered by a “SHOT” command in the protocol. When this command is encountered, the system maintains the current temperature and sends a request to the optic server for image acquisition. After the final cycle, the protocol proceeds to a final extension step (Label 5 at 72 °C for 3 min).

Each protocol row *A*(*i*) is defined as a triple (*L*, *T*, *D*), where

*L* is the label (or command, such as “SHOT” or ”GOTO”),*T* is the target temperature, and*D* is the duration at that temperature.

Then the protocol can be regarded as a sequence of actions as follows:*Q* = {*A*(*i*)|*A*(*i*) = (*L*, *T*, *D*), *i* = 1, …, *n*}(1)

The PCR controller processes each row as shown in Figure 5. For standard temperature commands, the controller sends a target temperature *Tc* to the PCR MCU and waits until the temperature is within a specified tolerance. Once the target is reached, a countdown begins for the specified duration.

If the label is “GOTO”, the controller decreases the cycle counter and jumps back to the indicated label. If the label is “SHOT”, it sends a fluorescence capture request to the optic server and waits until the operation is complete.

To maintain the target temperature, the PCR MCU employs a proportional–integral–derivative (PID) control algorithm. The target temperature accuracy was defined as ±0.1 °C, and this value was used as the allowable deviation criterion during performance evaluation. This margin was consistently achieved throughout repeated thermal cycling experiments. Communication between the host and PCR MCU uses the USB HID protocol with a 50 ms polling interval. During each cycle, the PCR controller sends the current target temperature and PID parameters and receives measured temperature values. The mathematical formulation of the PID controller and control loop parameters are provided in Appendix A.

#### 2.3.3. Optic Server

During real-time PCR, the relative fluorescence unit (RFU) value is obtained at the end of each cycle’s extension phase via the “SHOT” command, as shown in Table 1. Figure 6 illustrates the procedure for processing the “SHOT” command.

When the PCR controller encounters a protocol row labeled “SHOT”, it sends periodic TCP socket requests—every 1 s—to the optic server to initiate fluorescence capture. The optic server then executes the Shot Thread, which handles RFU calculation. This thread activates the LED by sending a PWM signal to the optic MCU and waits until a frame with maximum exposure is captured. Image processing is then performed to calculate fluorescence intensity, and the result is normalized to a 16-bit RFU value, which is sent back to the PCR controller. The internal thread structure and RFU acquisition process of the Optic Server are detailed in Appendix A.

#### 2.3.4. Synchronization of Camera and Illumination

For point-of-care testing (POCT) devices, using industrial cameras with built-in strobe synchronization features is often impractical due to cost. The open-platform camera used in this study does not support hardware-based synchronization between illumination and image capture. Therefore, a software-based method was employed to ensure the image sensor is fully exposed to LED light during frame acquisition.

Figure 7 illustrates the timing sequence used by the shot thread to acquire fully illuminated frames. Since fluorescence images are typically dim, the exposure time exceeds the frame time, making the exposure effectively span the entire frame duration. For example, if the shot thread turns on the LED at the start of frame *f*_0_, the resulting image transferred to the camera buffer may only be partially exposed (I_0_ duration). The next frame *f*_1_, however, is fully exposed to the LED and thus contains the desired fluorescence data.

To ensure this, the LED remains on until the end of frame *f*_1_, and the system relies on sleep delays to manage timing. However, this process is complicated by the asynchronous nature of the shot thread, camera thread, and frame transfer timings.

To determine the minimum reliable sleep time, the system conducted experiments in which fluorescence images were captured every 10 ms while the LED remained continuously on. The average brightness of each frame was measured, and the trend in brightness over time was analyzed. As shown in Figure 8, this experiment was repeated 100 times for each exposure time, and the results were overlaid to visualize consistency and identify when the brightness stabilized. The start time at which the intensity became stable was recorded for each exposure setting and is summarized in Table 2.

The results indicate that a sleep duration approximately three times the exposure time is sufficient (except at 0.0625 s). The shortest reliable exposure time that provided usable fluorescence data was 0.125 s. Thus, for consistency and reliability, the sleep duration was set to three times the exposure time in this system.

#### 2.3.5. RFU Calculation

To ensure accurate quantification of fluorescence signals, it is critical to precisely identify the region of interest (ROI) corresponding to the reaction chamber. Figure 9a shows a top-view image of the PCR chip filled with reagent, highlighting the polycarbonate chamber positioned above a thermal spreading copper pad on the PCB. Figure 9b shows a fluorescence image captured from a chip filled with FAM dye. The tight ROI, shown in white, was manually defined using ImageJ 1.54e to closely fit the chamber outline. While this tight ROI provides high signal-to-noise ratio (SNR) when well-aligned, the chamber position varies slightly between insertions (experimental variation) and between devices (device variation), making static ROI definitions unreliable.

To address this, we introduced a method to dynamically locate the ROI based on a position marker extracted from the image itself. Instead of using additional printed markers—which would increase chip manufacturing cost—we leveraged the nickel-coated copper pad located directly behind the chamber. Although the nickel coating does not fluoresce, its exposed edges emit a strong signal, particularly at the top-right corner, which can be detected and used as a reference marker.

Figure 9c illustrates the method used to locate this corner. Within a predefined corner search area, horizontal and vertical projection profiles are computed. The upper edge of the copper pad creates a distinct peak in the horizontal profile (shown in the inset above), and the vertical edge shows a similarly sharp peak (inset right). The intersection of these peaks indicates the corner position, which is marked with a white circle in both Figure 9b,c.

Figure 9d demonstrates the importance of this correction. The red outline shows a tight ROI applied based on the average marker position from another chip, which is misaligned in this image due to device variation. The corrected ROI, shown in white, was relocated using the detected copper corner and now properly encloses the chamber area. This correction improves both the consistency and reliability of RFU calculations across different chips and devices.

Since the copper pad’s outline is not always clearly visible in every chip, a simple projection-based corner detection is not universally applicable. Therefore, to evaluate the variation more accurately, we selected chips in which the copper pad outline was clearly visible, filled them with FAM solution, and inserted them six times into two different devices. Fluorescence images were captured for each insertion, and the right upper corners were detected using the projection method. From these, experimental variation (within the same device) and device variation (between different devices) were measured and are summarized in Table 3.

The results indicate that device variation was significantly larger than experimental variation, particularly in the horizontal (x) direction. Notably, because the chips are inserted into the connector until fully seated, horizontal (x-direction) variation was greater than vertical (y-direction) variation, as the insertion depth is constrained but lateral alignment can vary across devices.

Based on this analysis, a device-specific average marker position was stored, and the ROI was defined relative to this marker. To compensate for potential misalignment and background inclusion, the originally tight ROI was expanded using the 3σ rule, resulting in a “dilated ROI” used for RFU calculation. From Table 3, the maximum standard deviation across both devices was σx = 3.1 and σ_y_ = 1.1. Applying the 3σ rule yields dilation margins of approximately 9.3 pixels (horizontal) and 3.3 pixels (vertical), which led us to adopt a structuring element of 10 × 4 pixels for the dilation. This enlarged ROI ensures robust fluorescence signal capture even in the presence of device-level positional variation.

Figure 10 illustrates fluorescence images of chips whose detected marker positions were the furthest from the average marker location for each device. Both tight ROI and dilated ROI, defined relative to the average marker, are overlaid for comparison. In the case of Device 1, the mean absolute deviation (MAD) in the horizontal direction was 6 pixels. As shown in Figure 10a, the tight ROI fails to fully capture the fluorescent chamber area—missing part of the right side and including background on the left. In contrast, as shown in Figure 10b, the dilated ROI compensates for this offset, successfully covering the full chamber area while also including more background on the right side. For Device 2, the MAD was smaller, resulting in better alignment between the tight ROI and the actual chamber, as seen in Figure 10c. However, upon close inspection, part of the fluorescent chamber slightly exceeds the tight ROI on the left, while some background appears on the right. Figure 10d shows that the dilated ROI successfully recovers the left-side fluorescence region, though it also expands into the background region on the right.

Note that the ROI used during PCR runs is not generated dynamically but is created during a one-time device calibration stage using a reference chip loaded with FAM dye. The tight ROI is manually defined using ImageJ and then expanded based on measured variation across devices to form a robust dilated ROI. This pre-defined, device-specific ROI is stored in the Optic Server and applied in real time during each SHOT acquisition without additional fitting or adjustment.

Although alignment marks could be added to the mold to aid ROI alignment, we found this unnecessary due to the effectiveness of the calibration-based approach. Including such markers would have required structural modification and added manufacturing cost, whereas the current method enables accurate and reproducible ROI localization entirely through software.

### 2.4. RFU Curve Analysis

The RFU values measured at each PCR cycle form an amplification curve, as shown in Figure 11. A typical curve can be divided into four distinct phases: baseline, exponential, linear, and plateau.

The baseline phase corresponds to the early cycles where the fluorescence signal remains flat, reflecting only background noise.The exponential phase represents rapid amplification of DNA, with a sharp increase in fluorescence signal.The linear phase follows, showing a slower but steady increase.Finally, the plateau phase occurs when reagents become limiting and the fluorescence signal saturates.

The cycle threshold (Ct) is determined as the PCR cycle where the fluorescence signal first exceeds a predefined signal threshold. This signal threshold is typically fixed across experiments to ensure consistency in Ct detection. Ct is commonly used to determine the presence of target nucleic acids: if the Ct is lower than a preset decision threshold, the sample is classified as positive, indicating potential infection; otherwise, it is considered negative.

Accurate identification of the Ct point requires proper correction of background fluorescence signal variations during the baseline phase. In practice, the baseline segment can be determined in two main ways:By specifying a fixed cycle range, such as cycles 3–15 orBy dynamically identifying the start point of the exponential phase (SPE) and treating preceding cycles as baseline [1,37].

To correct for background variation, this study evaluated four baseline correction methods:

Fixed-Mean and SPE-Mean: subtracting the mean RFU of the baseline region.Fixed-Linear and SPE-Linear: subtracting a linear regression fitted to the baseline region.

To determine the most robust method, amplification curves were generated from repeated experiments using identical concentrations of standard DNA. The Ct variance was compared across the different correction strategies, and the method yielding the lowest Ct variance was selected as the optimal baseline correction technique for this system.

### 2.5. User Interface

The user interface (UI) of the proposed system was developed in Visual C++ on the host PC. As shown in Figure 12, the main UI is composed of six control groups:Connection group—located at the top, it manages system connectivity.RFU graph area—displays real-time RFU signals during the PCR run.Device status group—shows the current temperature of the PCR chip and the estimated remaining time.CT values group—displays the calculated Ct values for each fluorescence channel after PCR completion.Protocol group—allows users to run or stop protocols using the Start/Stop button and access protocol setup and editing via the Setup button.Result group—shows the positive/negative result for each target based on Ct thresholds.

When the Setup button is clicked, the Setup UI (Figure 13a) is opened. This includes the following:PCR Constants group—allows setting of PID parameters used for thermal cycling, mainly during system tuning and development.Protocol Manager group—opens the Protocol Editor UI (Figure 13b) for adding, modifying, or deleting protocols.Test History group—displays a list of previous experiments for reference and tracking.

In the Protocol Editor UI, each fluorescence channel is configured in the Filter group, which contains three input boxes for the following:Target DNA abbreviation,Positive/negative decision threshold (cycle number), andCt signal threshold.

The bottom part of the UI (Protocol group) contains controls for editing the thermal cycling protocol.

The performance of the proposed system was experimentally evaluated in terms of thermal control accuracy, fluorescence detection quality, and DNA quantification repeatability, as described below.

## 3. Results

### 3.1. Temperature Control Performance

To evaluate the temperature control performance of the PCR chip, 35 µL of distilled water was injected, and a thermal cycling profile consisting of five repetitions of 60 °C for 60 s and 95 °C for 60 s was performed.

As shown in Figure 14, the temperature profile demonstrated a rapid heating rate of 8.0 °C/s and a cooling rate of –9.3 °C/s on average (see Table 4). It is important to note that the reported temperature accuracy refers to measurements obtained from the NTC thermistor mounted on the PCB adjacent to the heating element. This sensor is used as the control reference in the PID feedback loop. While the actual reagent temperature may vary slightly due to thermal lag or material interfaces, the thermistor reading reflects the precision of the control system, which is critical for ensuring consistent PCR performance.

Steady-state performance was assessed by computing the mean error and standard deviation using the final 30 temperature samples of each step. As summarized in Table 5, the steady-state temperature error remained within ±0.1 °C, and standard deviations were all below 0.14 °C, indicating stable and precise thermal control.

### 3.2. Fluorescence Detection Performance

To determine optimal camera parameters for maximizing dynamic range in fluorescence detection, two sets of samples were prepared:1 pM/µL FAM solution to emulate plateau-level amplification andDistilled water to represent the baseline (no amplification).

The FAM solution served as a positive fluorescence control to emulate high signal intensity, while the distilled water acted as a negative control representing background noise. Both solutions were used to assess signal-to-noise ratio and optimize camera gain and exposure time. Each solution was loaded into five PCR chips (35 µL each), and images were captured using camera exposure times of 0.125, 0.25, and 0.5 s. A 0.5 s exposure provided the highest dynamic range without saturation.

Camera gain was also evaluated at levels of 0 and 50.

Figure 15 shows the raw fluorescence images of PCR chips filled with DW and FAM solution under both gain conditions. Figure 16 presents the cumulative histograms of pixel intensities for the corresponding FAM images, reflecting the intensity distribution within the defined ROI. In both gain settings, no noticeable pixel saturation was observed in the FAM images.

Table 6 summarizes the average fluorescence intensity within the ROI for each chip. Relative gain in the table was defined as (FAM−DW)/FAM. At gain 50, the relative gain improved from 6.4 to 7.3, representing a 22% enhancement, without causing pixel saturation. Based on these results, gain was set to 30 for subsequent DNA amplification experiments to ensure both adequate brightness and safe margin from saturation.

### 3.3. DNA Amplification and Quantitative Performance

The system was validated using a 35 µL PCR reaction containing 10^3^ copies of *Chlamydia trachomatis* DNA. The reaction mixture consisted of 5.25 µL of template DNA, 17.5 µL of commercial master mix, 8.75 µL of primer mix (0.28 pmol/µL), and 3.5 µL of double-distilled water (DDW).

The thermal cycling protocol is listed in Table 7, with 40 total cycles and fluorescence image acquisition performed after each extension step (60 °C) with 0.5 s exposure and gain of 30.

As shown in Figure 17, time-lapse fluorescence images of the PCR chip chamber were captured after each cycle under consistent imaging conditions. While early-cycle images appeared uniformly dark, a gradual increase in fluorescence signal became visually noticeable beginning around cycle 26, indicating the onset of exponential DNA amplification. This visual progression aligns with the quantitative amplification curve and supports the system’s ability to detect fluorescence increases corresponding to DNA replication.

Figure 18 presents raw amplification curves from 12 replicate tests using the same DNA concentration, and Figure 19 shows the results after baseline correction using the fixed-mean method.

Although ideal Ct values should be identical across replicates, small variations are inevitable. To evaluate reproducibility and optimize correction, four baseline correction methods were tested: fixed-mean, fixed-linear, SPE-mean and SPE-linear.

For each method, Ct standard deviation was analyzed over varying signal threshold levels. Figure 20 shows the minimum Ct standard deviation achieved at each log-threshold value. Across all methods, mean-based correction consistently outperformed linear regression, and the fixed-mean method yielded the lowest variation at σ ≈ 0.3 cycles, demonstrating excellent repeatability.

Overall, the system successfully demonstrated accurate temperature control, high-quality fluorescence acquisition, and consistent DNA amplification results. These findings are further discussed in terms of applicability, limitations, and future improvements in the next section.

## 4. Discussion

This study presents a low-cost, compact real-time PCR system suitable for point-of-care diagnostic applications in resource-limited settings. The PCR chip was fabricated using mass-producible PCB substrates and widely available materials such as medical-grade tape and polycarbonate, offering a cost-effective platform.

The thermal cycling system demonstrated excellent performance, achieving an average heating rate of 8 °C/s and cooling rate of −9 °C/s, with steady-state error within 0.1 °C, ensuring accurate temperature control.

For fluorescence detection, a CMOS-based open-platform camera was integrated using a slanted illumination setup to reduce system cost and size. The reduced lens and filter sizes further lowered component costs. As open-platform cameras lack strobe sync capabilities, the exposure timing for fluorescence capture was experimentally characterized. The results showed that keeping the LED on for three times the exposure duration was sufficient to obtain fully illuminated frames.

Fluorescence ROI selection was refined using device-specific marker detection and ROI dilation to account for both device and experimental variation. Compared to conventional photodiodes, this approach significantly enhances the SNR of fluorescence signals.

Finally, the system’s quantitative reliability was confirmed using standard DNA samples. With proper baseline correction and threshold optimization, the system demonstrated a Ct standard deviation under 0.3 cycles, indicating high reproducibility. Given its small footprint and low cost, the system is well-suited for on-site diagnostic use.

Compared to our previous works, this study represents a significant step toward practical POCT deployment. In [34], we demonstrated fluorescence detection using a DSLR camera, while [35] employed classical photodiode-based detection on a PCB-integrated chip. In [36], we validated four-color multiplex imaging using an open-platform camera and filter wheel, but the system remained limited to feasibility testing under controlled conditions. In contrast, the present study fully integrates all critical components—including thermal control, fluorescence detection, and image-based ROI tracking—into a low-cost, compact, and reproducible system. It is also the first among our works to provide quantitative thermal performance validation (±0.1 °C), implement camera synchronization strategies for real-time imaging, and simplify the system architecture to a one-color format optimized for POCT. Collectively, these features advance the system from conceptual feasibility to practical deployability in real-world POCT environments.

During experiments, the system also enabled easy visual monitoring of reagent leakage or injection failures via the integrated camera. This helped identify occasional leakage near the inlet during thermal cycling, suggesting that future improvements should focus on redesigning the inlet cap material and geometry.

Although the heater pattern was designed empirically without simulation-based optimization, the current layout provided adequate spatial thermal distribution, as evidenced by the measured ±0.1 °C temperature control accuracy and Ct variability below 0.3 cycles in repeated tests. In future work, thermal modeling and finite element analysis may be employed to refine the heater geometry for enhanced uniformity, lower power consumption, and faster thermal response.

If future mass production introduces increased device-level variation due to ultra-low-cost fabrication, automatic shape recognition algorithms may be explored as part of a factory-level ROI calibration process. However, in the current design, the use of a pre-characterized calibration chip enables sufficiently robust and repeatable alignment without the need for computationally intensive methods.

While the current study demonstrated amplification consistency using a standard concentration of Chlamydia trachomatis DNA, we recognize that a comprehensive evaluation of sensitivity and limit of detection (LoD) using serially diluted DNA samples remains unaddressed. This important validation will be pursued in future work to fully characterize the diagnostic capability of the proposed system.

To transition this system toward commercialization, further validation with various DNA concentrations and clinical samples is needed. Additionally, expanding the optical system for multiplex fluorescence detection and conducting clinical trials are necessary next steps. While direct clinical sensitivity has not yet been evaluated, the system achieved a Ct standard deviation below 0.3 cycles and ±0.1 °C thermal control accuracy under standard DNA conditions, which are comparable to the performance of conventional laboratory PCR systems. The current platform targets single-target detection of infectious diseases such as COVID-19, aiming to deliver RT-PCR-level sensitivity at a cost comparable to rapid antigen tests (RAT). With proper clinical validation and integration into a commercial-grade form factor, the system could be immediately applicable to frontline diagnostic use in resource-limited or high-throughput screening environments. Future clinical validation will include testing with a broader range of biological sample types—including crude specimens and matrix-rich clinical samples—as well as negative controls and benchmarking against established commercial PCR systems, to assess diagnostic specificity, robustness, and system compatibility across different sample types and primer configurations.

While the current prototype operated reliably under a 5 V 3 A power supply, a detailed analysis of system-wide power consumption has not yet been conducted. Future studies will quantify the thermal and electrical efficiency of each module to support optimization for portable or battery-operated applications.

## 5. Conclusions

We developed a compact and cost-effective real-time PCR system that integrates a PCB-based disposable chip, rapid thermal cycling, and CMOS camera-based fluorescence detection. The system demonstrated fast and accurate thermal control (±0.1 °C), optimized fluorescence imaging without saturation, and robust signal acquisition through device-specific ROI tracking. Quantitative validation using standard DNA samples showed high reproducibility, with a Ct variation below 0.3 cycles. These features position the system as a strong candidate for point-of-care testing (POCT), particularly in resource-limited environments where cost, size, and simplicity are critical.

While the current system supports only single-channel detection, this configuration aligns with single-target diagnostic needs similar to rapid antigen tests. Future work will focus on expanding the optical system for multiplex detection, integrating thermal simulation to optimize heater geometry, and conducting comprehensive clinical validation with crude and matrix-rich samples. Additional directions include developing factory-level ROI auto-alignment algorithms and analyzing system-wide power consumption for portable, battery-operated deployment. With continued refinement, the system may serve as a scalable platform for decentralized diagnostics in rural clinics, outbreak zones, and mobile health units.

## Figures and Tables

**Figure 1 sensors-25-03159-f001:**
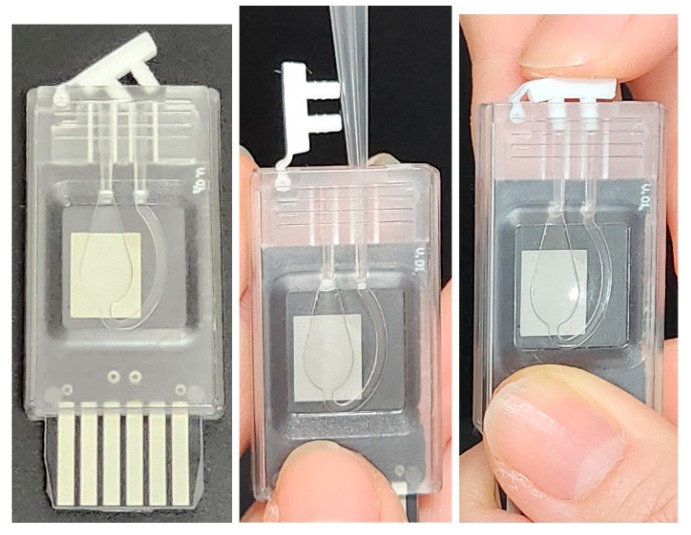
PCR chip and reagent injection procedure. The chip dimensions are approximately 22 mm (width) × 50 mm (length) × 5 mm (thickness).

**Figure 2 sensors-25-03159-f002:**
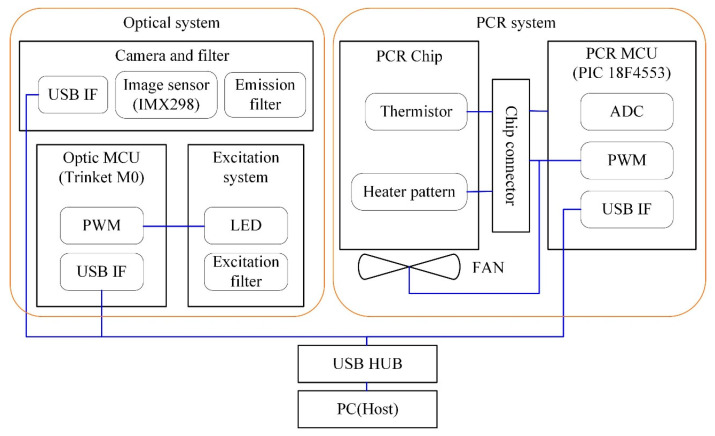
Hardware functional block diagram.

**Figure 3 sensors-25-03159-f003:**
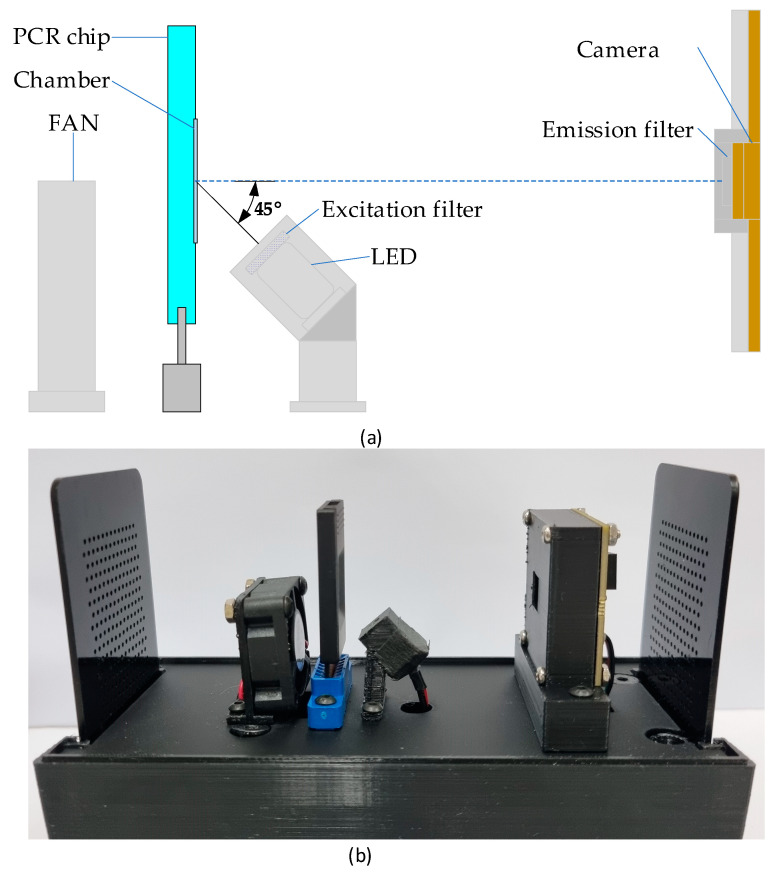
The system block diagram (**a**) and its implementation picture (**b**). The assembled system measures approximately 80 mm (width) × 155 mm (length) × 130 mm (height).

**Figure 4 sensors-25-03159-f004:**
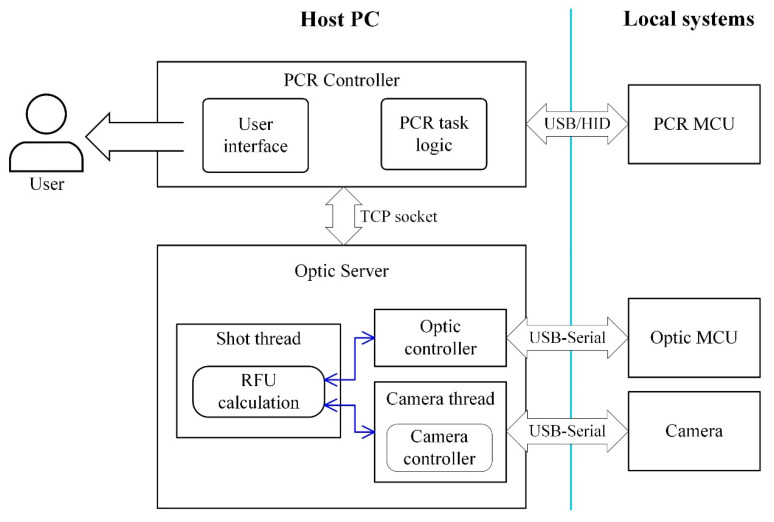
Software architecture.

**Figure 5 sensors-25-03159-f005:**
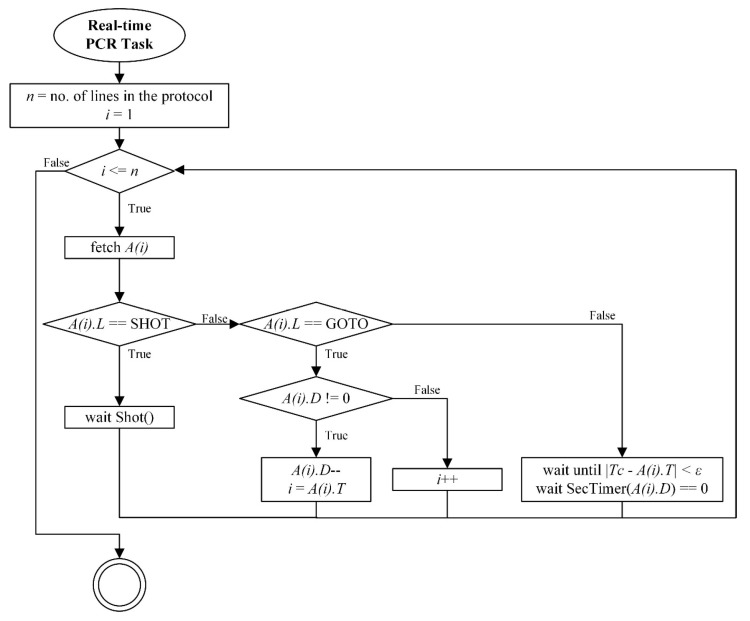
Real-time PCR protocol processing flow diagram.

**Figure 6 sensors-25-03159-f006:**
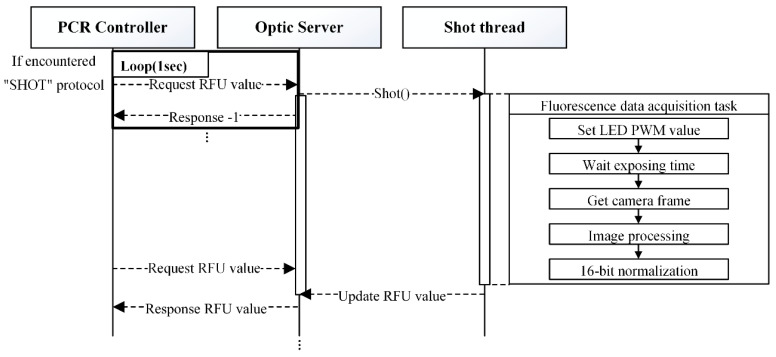
“SHOT” command processing.

**Figure 7 sensors-25-03159-f007:**
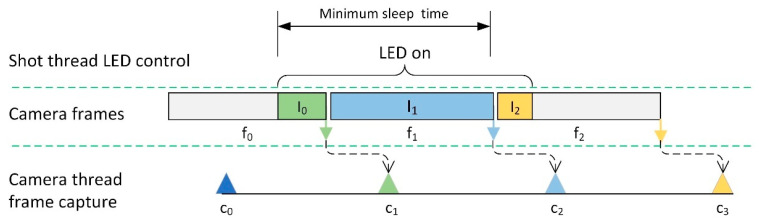
Timing sequence for acquiring a fully illuminated fluorescence frame using asynchronous LED activation and camera capture. The LED is turned on by the shot thread (top layer) and remains on long enough to ensure the camera captures a frame (e.g., *f*_1_) that is fully exposed to the excitation light. Due to the asynchronous nature of the shot thread, camera thread, and frame transfer, the LED must remain on for a sufficient duration to ensure reliable image acquisition.

**Figure 8 sensors-25-03159-f008:**
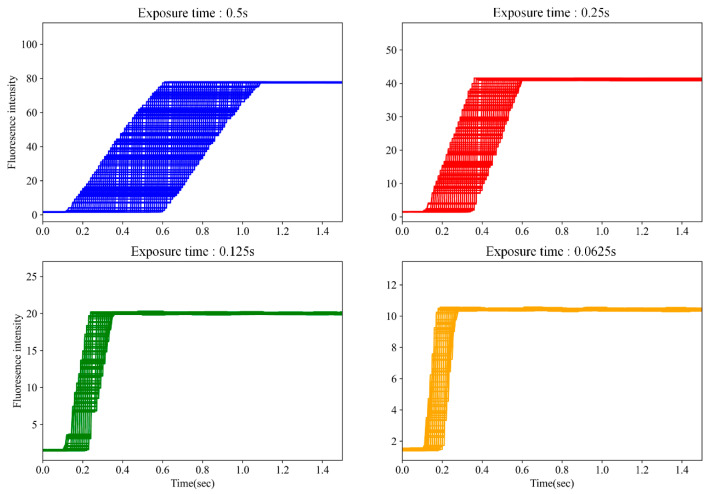
Average fluorescence intensity trends across 100 repeated image sequence acquisitions for different exposure times. Each curve shows how the mean brightness evolves over time after LED activation. The vertical axis represents relative intensity, and the horizontal axis represents time. The point at which brightness stabilizes was used to determine the minimum required LED-on duration for each exposure condition.

**Figure 9 sensors-25-03159-f009:**
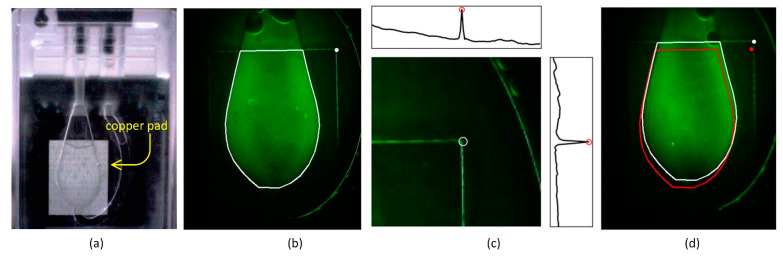
ROI localization using copper pad edge detection for reliable fluorescence quantification. (**a**) Image of a PCR chip filled with reagent, showing the chamber and the underlying nickel-coated copper pad used for thermal spreading. (**b**) Fluorescence image of a chip filled with FAM dye, with the manually defined tight ROI (white outline) and the detected marker point (white circle). (**c**) Method for detecting the right upper corner of the copper pad using projection peaks in a predefined search area. Insets show distinct horizontal and vertical intensity peaks used to locate the corner, marked with a white circle. (**d**) Comparison between a misaligned tight ROI (red outline) based on average marker position and a corrected ROI (white outline) relocated using the detected corner. The corrected ROI better encloses the chamber region, reducing positional error across devices.

**Figure 10 sensors-25-03159-f010:**
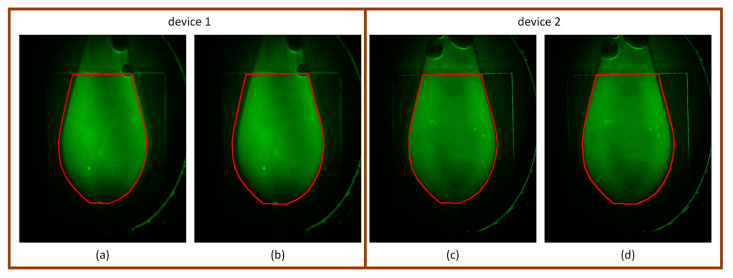
Comparison of tight and dilated ROIs on fluorescence images with large marker deviation. (**a**,**b**) Device 1: The tight ROI fails to fully capture the fluorescent chamber due to horizontal misalignment, whereas the dilated ROI successfully covers the entire chamber region. (**c**,**d**) Device 2: Although the tight ROI is relatively well-aligned, minor misalignment is still visible. The dilated ROI corrects this with a slight trade-off of including additional background signal.

**Figure 11 sensors-25-03159-f011:**
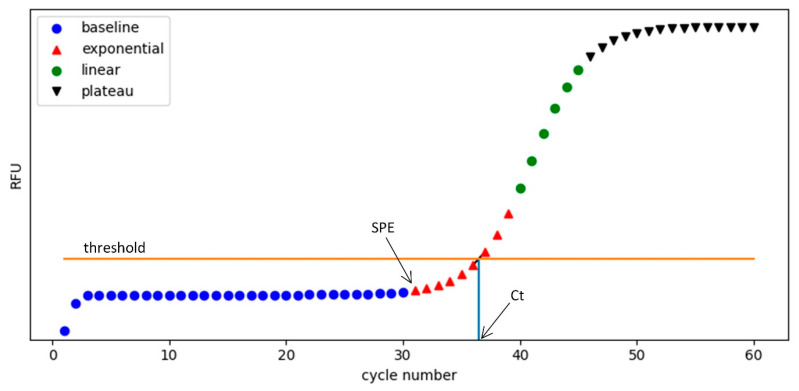
Characteristic phases of a real-time PCR amplification curve. The curve is divided into four segments: baseline (blue circles), exponential (red triangles), linear (green circles), and plateau (black inverted triangles). The horizontal orange line represents the signal threshold level for Ct determination. The start point of the exponential phase (SPE) marks the onset of rapid signal increase, while the cycle threshold (Ct) is the point where the fluorescence curve crosses the threshold. The Ct value is used as a diagnostic criterion: samples with Ct values lower than a preset decision threshold are considered positive.

**Figure 12 sensors-25-03159-f012:**
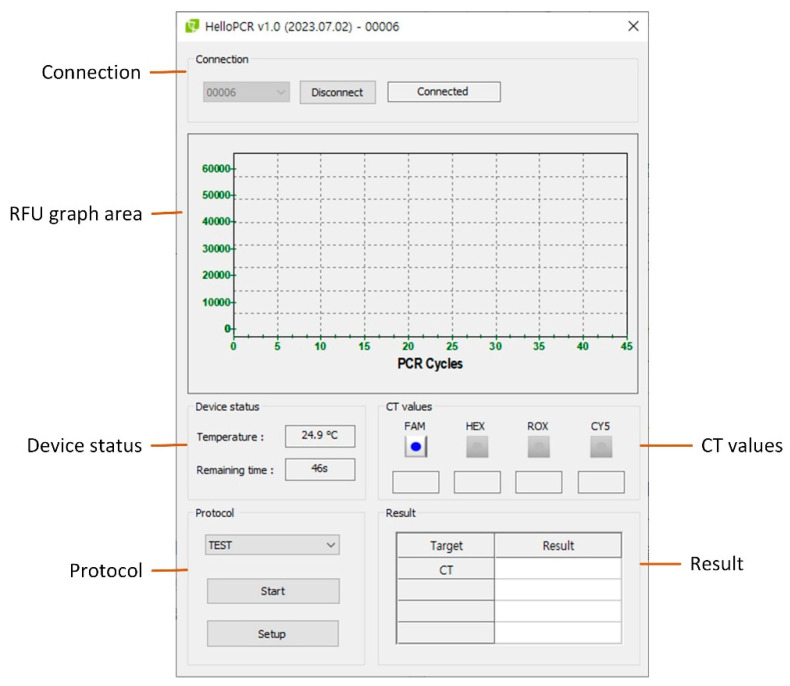
Main user interface of the PCR system. The UI contains six main control groups: (1) Connection, (2) RFU graph area, (3) Device status, (4) CT values, (5) Protocol control, and (6) Result panel.

**Figure 13 sensors-25-03159-f013:**
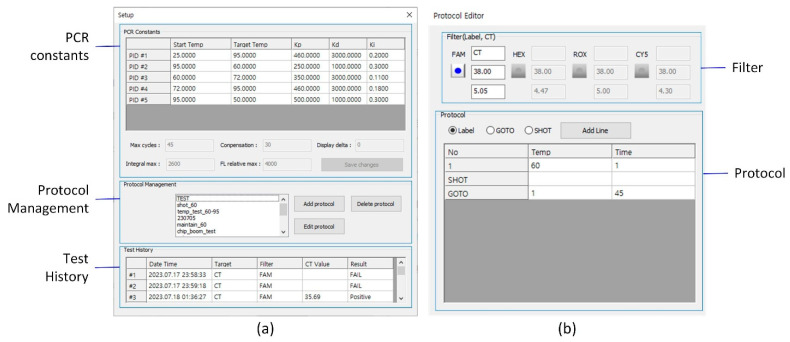
(**a**) Setup UI for configuring PCR constants, managing protocols, and accessing test history. (**b**) Protocol Editor UI for defining thermal cycling parameters and fluorescence channel settings, including the decision threshold for result classification and the signal threshold for Ct determination.

**Figure 14 sensors-25-03159-f014:**
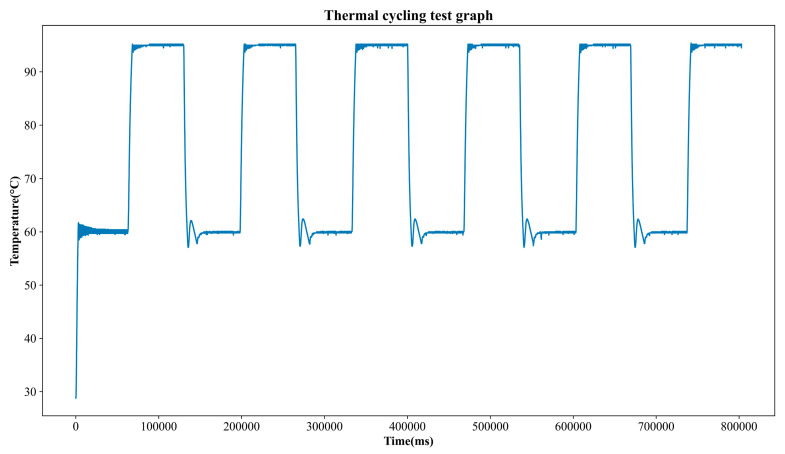
Thermal cycling test temperature profile.

**Figure 15 sensors-25-03159-f015:**
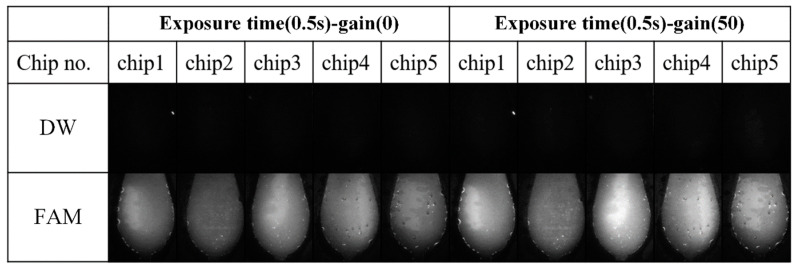
Fluorescence images of PCR chips filled with distilled water (DW) and 1 pM/µL FAM solution, captured at camera gain settings of 0 and 50. The images illustrate the contrast between background and amplified-like fluorescence signals, with no saturation observed at either gain level.

**Figure 16 sensors-25-03159-f016:**
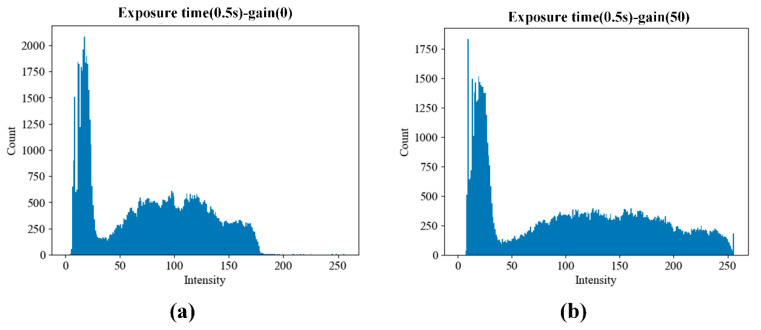
Cumulative histograms of pixel intensity within the region of interest (ROI) from PCR chips filled with 1 pM/µL FAM solution. Data were acquired using five replicate chips under two gain conditions: (**a**) gain = 0 and (**b**) gain = 50. The results confirm that both settings avoid pixel saturation while allowing sufficient signal differentiation for downstream quantification.

**Figure 17 sensors-25-03159-f017:**
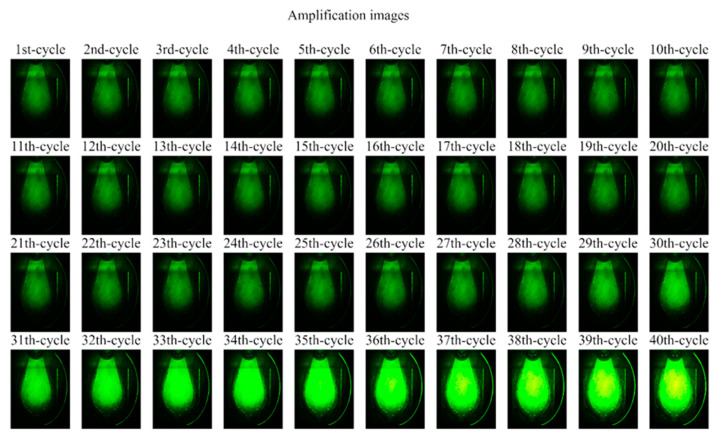
Time-lapse fluorescence images of the PCR chip chamber acquired at each PCR cycle. Images were captured with a camera exposure time of 0.5 s and gain of 30 after the extension step at 60 °C. Brightness and contrast were enhanced for visibility. Increased fluorescence intensity became noticeable starting from cycle 26.

**Figure 18 sensors-25-03159-f018:**
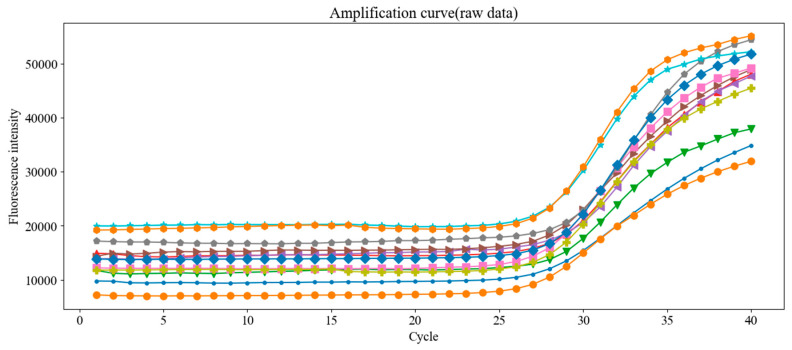
Raw fluorescence amplification curves from 12 replicate PCR runs using the same concentration of *Chlamydia trachomatis* DNA standard. Despite using identical samples, small variations in Ct values were observed, emphasizing the need for proper baseline correction.

**Figure 19 sensors-25-03159-f019:**
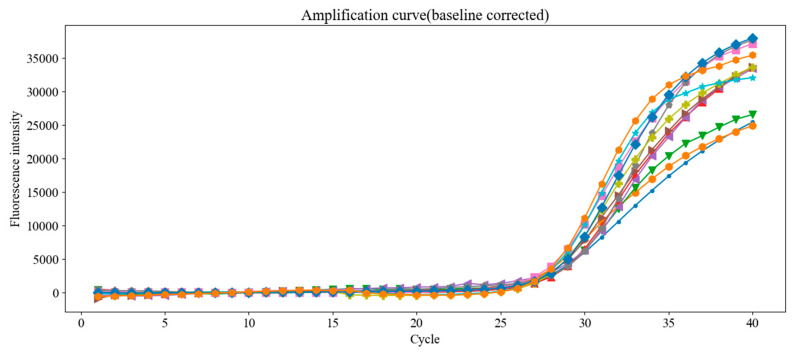
Fluorescence amplification curves after applying fixed-mean baseline correction to the data shown in Figure 18. The correction improves alignment and consistency of the curves, reducing Ct variability across replicate tests.

**Figure 20 sensors-25-03159-f020:**
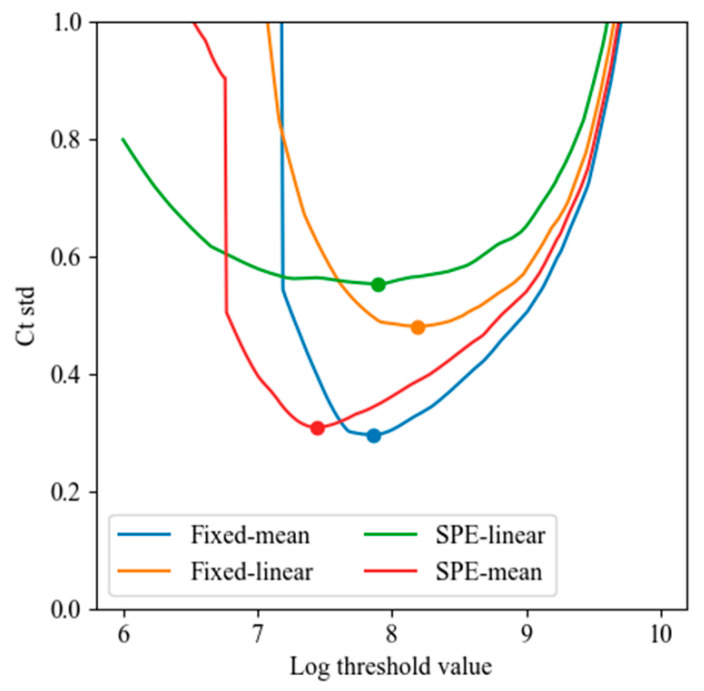
Minimum Ct standard deviation obtained by varying the threshold value (log scale) for each of the four baseline correction methods: fixed-mean, fixed-linear, SPE-mean, and SPE-linear. Mean-based methods outperformed linear regression methods, with the fixed-mean method achieving the lowest Ct variability (σ < 0.3 cycles).

**Table 1 sensors-25-03159-t001:** An example of real-time PCR protocol.

Label	Temperature (°C)	Duration (s)
1	95	300
2	95	30
3	55	30
4	72	30
SHOT	-	-
GOTO	2	34
5	72	180

**Table 2 sensors-25-03159-t002:** Minimum LED-on time required to obtain stable fluorescence intensity, based on 100 repeated measurements per exposure time. The stable start time indicates the earliest point where signal intensity no longer increases. The ratio shows the required LED-on duration relative to the camera exposure time.

Exposure Time (s)	Stable Start Time (s)	Ratio
0.5	1.18	2.4
0.25	0.6	2.4
0.125	0.36	2.9
0.0625	0.28	4.5

**Table 3 sensors-25-03159-t003:** Experimental and device-level deviations in copper pad corner location (in pixels). Data were collected from six insertions per device using chips with clearly visible copper outlines. “Range” refers to the maximum deviation observed, “MAD” is the mean absolute deviation from the device-specific mean, and σ indicates the standard deviation. Results show that device-to-device variation is significantly greater than experimental variation.

	x-Range	y-Range	x-MAD	x-MAD	σ_x_	σ_y_
device 1	9	3	6.0	2.2	3.1	1.1
device 2	5	1	2.7	0.7	1.8	0.5
device 1 & 2	15	21	9.2	11.3	4.1	9.1

**Table 4 sensors-25-03159-t004:** Temperature heating and cooling rate (°C/s).

Cycle	1	2	3	4	5	mean	std
Heating rate	7.6	7.9	8.0	8.1	8.1	8.0	0.2
Cooling rate	−9.3	−9.5	−9.4	−9.2	−9.3	−9.3	0.1

**Table 5 sensors-25-03159-t005:** Mean error and the standard deviation at the steady state (°C).

Cycle		1	2	3	4	5
95	mean	0.04	0.05	0.05	0.05	0.05
	std	0.11	0.11	0.12	0.11	0.11
60	mean	−0.06	−0.06	−0.07	−0.09	−0.07
	std	0.10	0.10	0.10	0.14	0.10

**Table 6 sensors-25-03159-t006:** ROI average intensity values for DW and FAM samples across five PCR chips at camera gains of 0 and 50. Relative gain (FAM−DW)/FAM quantifies signal enhancement due to amplification.

Camera Gain		chip 1	chip 2	chip 3	chip 4	chip 5	mean
0	DW	11.6	11.4	11.6	11.8	12.5	11.8
	FAM	82.8	69.0	103.1	96.9	81.8	86.7
	Relative gain	6.1	5.1	7.9	7.2	5.6	6.4
50	DW	14.0	13.7	14.0	14.4	15.0	14.2
	FAM	113.5	90.1	143.9	132.1	112.2	118.4
	Relative gain	7.1	5.6	9.3	8.2	6.4	7.3

**Table 7 sensors-25-03159-t007:** Thermal cycling protocol used for the amplification of *Chlamydia trachomatis* DNA. Labels 1 and 2 correspond to initial activation and denaturation steps, followed by 39 cycles of denaturation (Label 3), annealing/extension (Label 4), and fluorescence capture (SHOT). The GOTO command loops back to Label 3 for repeated cycling.

Label	Temperature (°C)	Duration (s)
1	50	120
2	95	600
3	95	15
4	60	60
SHOT	-	-
GOTO	3	39

## Data Availability

The original contributions presented in this study are included in the article. Further inquiries can be directed to the corresponding authors.

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
