# Peer review of "A Compact Real-Time PCR System for Point-of-Care Detection Using a PCB-Based Disposable Chip and Open-Platform CMOS Camera"

_sensors, 2025, doi:10.3390/s25103159_

Round 1
Reviewer 1 Report
Comments and Suggestions for Authors
Dear Authors
The manuscript introduces an innovative, well-designed, and functional real-time PCR system, specifically developed for point-of-care testing (POCT). The writing and organization of the manuscript are generally strong, making it suitable for publication in a peer-reviewed journal."
- "The manuscript effectively details how the use of a PCB-based disposable chip and an open-platform CMOS camera presents an innovative strategy to decrease both the system's cost and complexity."
- "The manuscript offers the necessary technical details of the system's design, covering aspects such as the PCR chip, hardware architecture, and software components."
- "The descriptions of the thermal cycling system, the detection system, and the software synchronization are clear and are well-supported by the inclusion of relevant figures and diagrams."
- "The validation of the system's performance using Chlamydia trachomatis DNA is appropriate, and the results presented are convincing."
- "The manuscript is organized in a way that makes it easy for the reader to follow the presented information."
- "The language used in the manuscript is clear and concise."
- "The figures and tables included in the manuscript are well-described and used effectively to support the text."
"The manuscript would benefit from some modifications, including:"
- "Correction of the error in the figure number and citation (Figure 2)."
- "Enhancement of the discussion section by incorporating a comparison between the developed system and existing POCT real-time PCR systems, which would add further value to the manuscript."
- "Please clarify the multiplexing capabilities of this system.
Reviewer 2 Report
Comments and Suggestions for Authors
This manuscript touches an important monitoring tool related with the development of a PCR system for Point-of-Care Detection, however I note a several issues. This manuscript needs a strong reconfiguration to the MDPI Sensors requests.
- The authors should clearly highlight the main innovations compared to their previous work [34, 35, 36], in order to emphasize the novel contributions achieved in this study.
- Several materials are available for the fabrication of microfluidic devices. Why was polycarbonate selected as the material in this case? Further justification should be provided (main properties, biocompatibility, reproducibility, cost, among others).
- A brief justification must be provided for why the PCB is designed to be disposable. Could this design be reconsidered to allow for reusability, thereby avoiding unnecessary waste and more effectively supporting the stated low-cost objective?
- Do you consider your device to be miniaturized and portable for use in low-resource settings? The overall dimensions, weight, and estimated cost should be included to clarify these points. Scales must be added in figures (for example in figure 1 and 3).
- How was confirmed the accurate thermal control (±0.1°C)? Detail about this characterization should be included.
- More details about the fabrication process of the polycarbonate chamber must be provided. Additionally, the authors must add the dimensions of the chamber and channel.
- The authors should include a figure showing the heating pattern (including scale). Were numerical simulations performed to determine the geometry? How was the optimized geometry achieved?
- Do the authors consider a fan the most viable solution for a portable and miniaturized system? Please discuss its suitability compared to other alternatives, such as Peltier devices, particularly in terms of miniaturization.
- The authors must add detailed information about the electronic circuit and image acquisition hardware
- The authors must add references about all components (e.g. thermistor, fan, excitation and emission filter, lens, among others) used in the implementation. If possible, it was interesting and improved the manuscript if a total cost estimation could be included.
- Figure 2 must be corrected. In its current form, the figure is incorrectly numbered.
- What is the device's power consumption? How is it powered, and what is its autonomy?
- The manuscript lacks detail about several modules. It is suggested that a supplementary section be added with detailed information about the implementation of:
- Host-local architecture
- Optic Server
- Proportional–Integral–Derivative (PID) control algorithm
- There are repetitions in the manuscript:
- Line 243-245
- Line 584
- The acquired fluorescence images require further processing with ImageJ to fit the chamber outline. Is this processing performed sequentially during image acquisition, or only at the end? Please, clarify this point, especially when the authors indicate “Real-Time PCR System”.
- The authors use the detection of the edges of the nickel-coated copper pad for dynamically locate the ROI. Why not use alignment marks, which could be easily included into the chamber mold? Was the additional cost of the nickel-coated copper pad intended solely to facilitate alignment or also to enhance contrast in fluorescence image acquisition?
- Could a solution based on automatic shape recognition algorithms simplify the implementation?
- Are the sensitivities obtained comparable to those of existing systems? A brief comparison should be included in the manuscript.
- The technology could be used as a standard acceptable in the current healthy industry and clinical requirements? Please discuss its potential for further developments.
The English could be improved to more clearly express the research.
Reviewer 3 Report
Comments and Suggestions for Authors
This manuscript presents the design, development, and validation of a compact, cost-effective, real-time PCR platform designed for point-of-care testing (POCT). The study addresses a highly relevant challenge in molecular diagnostics by proposing a system that integrates a PCB-based thermal cycler and CMOS-based fluorescence detection, with promising potential for use in resource-limited settings.
The work is technically sound and presents encouraging results. However, the manuscript would benefit from several improvements to enhance clarity, scientific rigour, and completeness. The following recommendations are provided for the authors’ consideration:
- The manuscript requires a thorough English language revision. Several grammatical inconsistencies, awkward phrasings, and punctuation errors were observed. A professional proofreading or editing service is recommended to ensure clarity and fluency throughout the text.
- All Latin species names must be written in italics according to scientific conventions.
- All abbreviations should be defined in full at their first occurrence in the manuscript. Ensure consistency in their usage throughout the document.
- Please verify that all references cited in the text are correctly listed in the reference section and vice versa.
- There is a duplication of Figure 1; please revise the numbering throughout the manuscript to ensure each figure is uniquely and sequentially labelled.
- Figures 18 and 19 should be augmented with data labels, where appropriate, to enhance the clarity and interpretability of the results presented.
- FAM and DW Solutions - Please provide more detailed information on the composition, concentrations, and roles of the FAM dye and DW solutions used in the fluorescence detection experiments. Clarify whether these were controls, calibration agents, or parts of the test samples.
- The Results and Discussion sections would benefit from a detailed comparison with existing commercial or published results on PCR systems. Include cost, thermal ramp rates, Ct precision, size, and throughput benchmarks. Discuss how the presented system advances the field or addresses specific technological shortcomings.
- Although the system demonstrates strong technical capabilities, please consider including a brief discussion of limitations—such as possible challenges in multiplexing, sensitivity to environmental conditions, or integration with sample preparation steps.
- Conclusions - The statement “Fast and accurate thermal control (±0.1 °C error)” is repeated twice in the conclusion and should be corrected.
- Transform the conclusion from a bullet-point list into a coherent narrative that synthesizes the main achievements, addresses any observed limitations, highlights the work's real-world significance, and mentions potential applications and future perspectives, such as multiplexed detection, integration with microfluidics, or deployment in specific field settings (e.g., rural clinics or outbreak zones).
- The manuscript requires a thorough English language revision.
Reviewer 4 Report
Comments and Suggestions for Authors
- The manuscript presents an innovative approach to developing a cost effective and portable PCR system for point of care testing (POCT), which is particularly relevant for resource limited settings. The system has been thoroughly validated, demonstrating excellent repeatability with a cycle threshold (Ct) variation of less than 0.3 cycles.
- The manuscript is rich in technical detail, which, while valuable, may make it difficult for a broader audience to follow. A more structured and concise presentation would enhance the readability of the paper. Specifically, clearer figures and tables that summarize key performance metrics would help readers absorb the technical details more easily.
- Although the authors describe the system as low cost, the manuscript does not include an explicit breakdown of the component costs or a comparison with commercial PCR systems. Including such a comparison would strengthen the paper’s claims regarding the affordability of the system and provide a clearer picture of its economic viability.
- The manuscript does not include negative controls or a comparison with a gold standard commercial PCR system to benchmark the sensitivity and specificity of the proposed system. Including such data would provide a clearer understanding of the system’s performance in comparison to established technologies, strengthening the manuscript’s validity.
- Temperature Control Tolerance (Page 7, Lines 202-203): The manuscript mentions that "the controller sends a target temperature 𝑇𝑐 to the PCR MCU and waits until the temperature is within a specified tolerance." It would be helpful to explicitly define what this tolerance is to provide clarity to the reader.
- It would be beneficial to include additional biological testing data that demonstrate the system’s efficacy with different sample types. Furthermore, I suggest moving some of the more technical figures (e.g., block diagrams) to the supplementary materials to make the main text more accessible to readers without a technical background.
- The manuscript does not mention the specific primers and reagents used in the PCR experiments. Including this information is crucial for reproducibility and transparency, as well as for enabling readers to assess the system's performance under standardized conditions.
- I recommend that the authors evaluate the system using primers of varying lengths (e.g., short, medium, and long) to demonstrate its robustness and compatibility across different primer sizes.
- Figure 1 (Page 3, Line 106): shows a photographic image of the PCR chip. To improve clarity, I recommend including a 3D model or schematic of the chip that clearly labels the individual components. Adding dimensions to the diagram, and comparing the chip’s size to an everyday object (such as a coin), would help readers better grasp the scale of the system.
- Hardware Functional Block Diagram (Page 4, Line 132) is incorrectly presented as Figure 1.
- Figure 3 should include the total dimensions of the system block to give the reader a clearer understanding of the size and form factor of the device.
- Table 4 (Page 16, Line 464): the temperature heating and cooling rates are presented. It would be helpful to include both the mean and standard deviation values for these rates to give a better sense of the variability and reliability of the system’s performance.
- The figure legends are currently brief and lack sufficient detail. I recommend expanding the legends to provide more context and explanations to ensure that the figures are fully understood by readers without needing to reference the main text.
- Several minor grammatical errors are present throughout the manuscript. These should be addressed to improve readability and ensure the paper maintains a professional tone.
Round 2
Reviewer 2 Report
Comments and Suggestions for Authors
The authors have revised the manuscript in accordance with the previous review comments, effectively demonstrating the feasibility of the Real-Time PCR platform toward practical point-of-care deployment. This study offers a valuable tool for the rapid diagnosis of infectious diseases. I think the revised manuscript is now suitable for publication.
Reviewer 3 Report
Comments and Suggestions for Authors
The authors have adequately addressed most of the recommendations.
However, some references are listed in the reference section but not properly indicated in the text (for example, reference 37).
Comments on the Quality of English Language
The English language needs revision.
Reviewer 4 Report
Comments and Suggestions for Authors
The authors have satisfactorily addressed most of my concerns, and I find many of their responses reasonable. However, I strongly request the inclusion of a PCR amplification curve using serially diluted DNA samples to demonstrate the system’s performance across a range of DNA concentrations. This is essential for properly assessing its sensitivity and limit of detection. Data from a single high concentration DNA sample is insufficient for these purposes.
